

# Influence of sex and maturity state on trace elements content in liver and muscle of the Sciaenidae *Totoaba macdonaldi*

Lia C. Méndez-Rodríguez[1,*], Berenice Hernández-Aguilar[1],
Juan A. de Anda-Montañez[2], Eduardo F. Balart[2], Martha J. Román-Rodríguez[3]
and Tania Zenteno-Savín[1,*]

[1] Programa de Planeación Ambiental y Conservación, Centro de Investigaciones Biológicas del Noroeste, S.C., La Paz, Baja California Sur, México
[2] Programa de Ecología Pesquera, Centro de Investigaciones Biológicas del Noroeste, S.C., La Paz, Baja California Sur, México
[3] Comisión de Ecología y Desarrollo Sustentable del Estado de Sonora, San Luis Rio Colorado, Sonora, México
[*] These authors contributed equally to this work.

Corresponding author
Tania Zenteno-Savín,
tzenteno04@cibnor.mx

## ABSTRACT

**Background**. The fish *Totoaba macdonaldi* is endemic to the Upper Gulf of California. Its migratory movements involve sites with high levels of trace elements in the environment that can accumulate in tissues. In this study, lead (Pb), copper (Cu), cadmium (Cd), zinc (Zn) and iron (Fe) concentrations in male and female totoaba liver and muscle were quantified at various sexual maturity stages along the species' geographic distribution.

**Methods**. Generalized linear models were used to explore associations between trace element concentrations and season of the year, sex/maturity stage, and total fish length.

**Results**. No detectable Pb concentrations were recorded in liver or muscle; Cu, Cd, Zn and Fe contents in totoaba liver and muscle were typical of fish inhabiting areas with no contamination issues and are within international maximum permissible levels for human consumption. Variations in the content of Cd, Cu, Zn and Fe in liver of totoaba seem to be more related to the feeding and reproductive physiology of this species than as result of environmental exposure. Results suggest that consumption of totoaba muscle does not pose a public health risk. Furthermore, depending on the sex/maturity stage of totoaba, this fish's muscle may provide approximately 70% Cu, 60% Zn and 100% Fe of the recommended dietary reference intake.

## INTRODUCTION

*Totoaba macdonaldi* (Gilbert, 1890), commonly known as totoaba (*Page et al., 2013*), is an endemic fish found in the Gulf of California. Totoaba has a long life of up to 25 years (*Román Rodríguez & Hammann, 1997*) and late sexual maturity with an average size at first reproduction of 1,240 mm (*De Anda-Montañez et al., 2013*). It is a highly selective

piscivorous predator that feeds primarily on anchoveta (*Cetengraulis mysticetus*) and, to a lesser extent, on the silverside of the genus *Colpichthys* as well as several species of mollusks (*De Anda-Montañez et al., 2013*). Among the Sciaenidae inhabiting this region, *T. macdonaldi* displays the greatest size and weight, with up to two meters in length and weighing over 100 kg (*Guevara, 1990*; *Cisneros-Mata, Botsford & Quinn, 1997*; *Román Rodríguez & Hammann, 1997*). These attributes make it attractive for fishing. In fact, totoaba fishery was important in the first half of last century in the Gulf of California, yielding over 2,000 tons in 1942; however, in the 1970s catches decreased to a minimum and the Mexican government declared a permanent ban since 1975 (*Arvizu Martínez, 1987*; *Flanagan & Hendrickson, 1976*). In the last decade, techniques for the aquaculture (mariculture) of totoaba have been developed (*Rueda-López et al., 2011*; *True, Silva Loera & Castro Castro, 1997*) and, due to its organoleptic properties, totoaba has become a highly sought-after item in restaurants. Establishing base line trace elements concentration in muscle of totoaba is relevant to characterize and improve the knowledge about their biology.

Trace element content in organisms depends on the species' capacity to regulate metal (e.g., copper (Cu), zinc (Zn)) concentration and, within tissues, is associated to the specific tissue's function and enzymatic makeup (*Hamza-Chaffai et al., 1995*). Sex and reproductive stage have a significant effect in the concentrations of trace elements in marine organisms. In marine and freshwater female fish, liver Zn content linked to metallothionein increases during reproduction, potentially aiding vitellogenesis and the synthesis of estradiol-17$\beta$ (*Thompson et al., 2001*). The demand for Zn and Cu increases during embryonic and neonatal development (*Olsson, Haux & Förlin, 1987*; *Thompson et al., 2002*). During the annual reproductive cycle of the rainbow trout, *Salmo gairdneri* Cu levels in livers from male fishes were higher than those observed in the females and the Cu levels in males also showed larger fluctuations than females (*Olsson, Haux & Förlin, 1987*). The liver is the primary organ where trace elements are stored and physiologically regulated in fish (*Hamza-Chaffai et al., 1995*; *Roméo et al., 1999*). Variability in trace element concentrations is associated to increased accumulation with age, to the geographic location, and to the species' feeding ecology.

Marine organisms found near the coasts of industrial and agricultural centers, as well as those that feed in the upper levels of the food chain, have higher concentrations of trace elements and other contaminants (*O'Shea & Brownell, 1994*), although there are alternative suggestions (i.e., *Mathews & Fisher, 2009*). In the Gulf of California, trace element concentrations change due to natural phenomena, such as upwelling, and anthropogenic activities, including mining and agriculture. Cadmium is abundant in the Gulf of California and its concentrations are associated to upwelling processes (*Ruelas-Inzunza & Páez-Osuna, 2008*). Naturally occurring Cu deposits in Santa Rosalia, that also contain Zn, have been intermittently exploited since the 18th century (*Shumilin et al., 2000*). Sediment lead (Pb) concentrations in some urbanized areas along the coasts are relatively high (up to 89 ppm) and it has been suggested that these reflect anthropogenic activities (*Shumilin et al., 2001*). Adult totoabas reach the Upper Gulf of California to spawn from late winter to early spring in areas adjacent to the Colorado River delta, where they

 

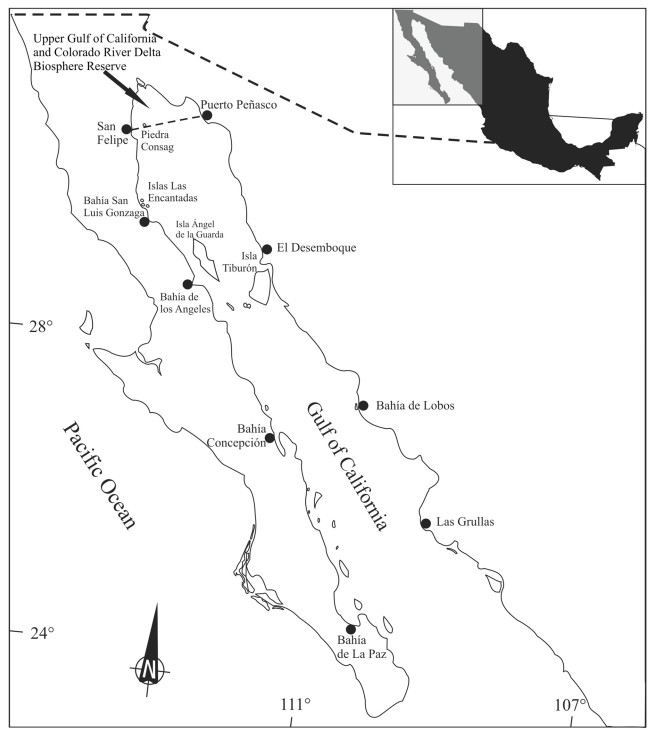

**Figure 1** *Totoaba macdonaldi* **study area in the Gulf of California.**

remain until late May-early June; then, they migrate southwards to the waters surrounding the large islands (Ángel de la Guarda and Tiburón Islands) (Fig. 1). Totoaba remain in this area until late fall (October), when they travel north along the coast of Sinaloa and Sonora to subsequently return to the Colorado River delta in January–February (*Arvizu & Chávez, 1972*; *Cisneros-Mata, Montemayor-Lopez & Roman-Rodriguez, 1995*). Therefore, this species' migration includes sites enriched by trace elements as a result of either natural conditions or anthropogenic inputs and areas where human activities, mostly agriculture, influence water quality (*Cadena-Cárdenas et al., 2009*).

Trace metals tend to accumulate in marine organisms that are included in the human diet. Some trace elements such as Zn, Cu, and iron (Fe) are essential elements involved in different metabolic processes; but others, such as Pb and cadmium (Cd) are non-essential elements that compete with essential elements for enzyme sites. Although less than 20% of Pb and Cd are assimilated in the human body, their half-life in human tissues is around 30 years (*EFSA, 2009*; *EFSA, 2010*). Even essential trace elements in excess can cause serious effects to both human and animal health. For example, high intake levels of Cu, Zn, and Pb have been related to Alzheimer's disease; Zn and Fe, with Parkinson's disease; Cd may induce kidney dysfunctions, osteomalacia, and reproductive deficiencies, among others (*Han et al., 2013*; *Manea et al., 2020*). Due to fish being an essential source of nutrients for humans, but potentially also a source of metals in concentrations that can cause health problems to the consumers (*Ruelas-Inzunza, Páez-Osuna & García-Flores, 2010*;
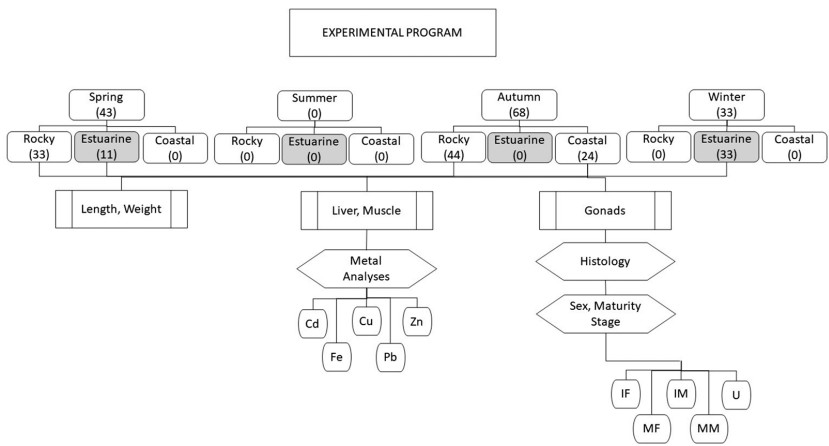

**Figure 2** Overview of experimental program.

*Zamora-Arellano et al., 2018*), it is necessary to perform health risk assessments to identify the adequacy of fish species for human consumption.

The objective of this study was to assess changes on the concentrations of essential elements (Cu, Zn, Fe) and toxic elements for humans (such as Cd and Pb) in relation to the sex/maturity stage, season, and habitat in liver and in the edible tissue (muscle) of totoaba. This information can provide support for totoaba aquaculture which can contribute to the protection of this species if it is no longer fished.

## MATERIAL & METHODS

### Overview of experimental program

An overview of the experimental program is shown in Fig. 2. Totoaba individuals ($n = 144$) were collected from the Upper Gulf of California in three types of habitats, (1) estuarine, (2) rocky, and (3) continental. Field trips were conducted to collect samples during all seasons (spring, summer, autumn, winter) along the species' distribution; however, fish were not found in summer in any of the sampled sites; in winter and spring totoaba were found in the spawning grounds (estuarine habitat), while specimens sampled in autumn were collected in non-spawning areas (coastal and rocky habitats) (*De Anda-Montañez et al., 2013*). From each totoaba caught, total length and weight were recorded, and liver ($n = 142$) and muscle ($n = 144$) samples, as well as gonads ($n = 144$), were collected. Based on the histological analysis of gonads and on gamete development and tissue characteristics sex and maturity stage were defined.

### Sample collection

Totoaba individuals were collected from the Upper Gulf of California (Fig. 1) under scientific collection permits issued by the Mexican government (SGPA/DGVS/02913/10, SGPA/DGVS/05508/11 and SGPA/DGVS/00039/13) based on the research protocol described in the project entitled "Health and Conservation status of the totoaba population (*Totoaba macdonaldi*) in the Gulf of California: a critically endangered species" (CONABIO:

FB1508/HK050/10; CONACYT 2011-01/165376; CIBNOR:137C/EP0.04). This study was performed following the demands, requirements and protocols that are required by the various institutions involved (SGPA/DGVS, CONABIO, CONACYT, CIBNOR). A total of 144 individuals were collected between May 2010 and April 2012. The organisms were collected in three types of habitats; (1) estuarine, which correspond to the area of the Upper Gulf of California and Colorado River Delta Biosphere Reserve, (2) rocky, refers to the area around the Islands, and (3) continental, which refers to the east coast of the Gulf of California in different seasons of the year along the species' distribution across the Gulf of California. Fishing was carried out on board 26-foot vessels with outboard motors. The fishing gears used included gillnets measuring 120 m long with a mesh size of 10 inches, and fishing rods with no. 5 and 6 hooks; shrimp heads, squid and/or comber were used as bait. The total length and weight of each totoaba caught were recorded. Liver (142) and muscle (144) samples were collected from each fish; samples were separately placed in plastic bags, stored on ice, and transported to the laboratory at CIBNOR for storage at −80 °C until analyzed. Gonads ($n = 144$) were also sampled and stored in Davidson's fixative for sex identification and maturity stage determination. Each individual fish was classified as follows, based on the histological analysis of gonads, immature female (IF), mature female (MF), immature male (IM), and mature male (MM). This classification was based on gamete development and the characteristics of the reproductive structures, such as follicles, germinal epithelium, and interfollicular connective tissue (*Tyler & Sumpter, 1996*). Fish in which gonads lacked characteristics allowing for the differentiation of ovaries or testicles were classified as undifferentiated (UN).

## Trace Elements Analysis

From each totoaba collected, 5 g of muscle and 5 g of liver were oven-dried separately at 70 °C and subsequently digested with nitric acid ($HNO_3$) and $H_2O_2$ in a microwave (Mars 5x, CEM, Matthew, NC, USA). After acid digestion, 1 mL of hydrochloric acid (HCl) was added to each sample and the volume was brought to 50 mL with deionized water (*Cadena-Cárdenas et al., 2009*; *Barrera-García et al., 2012*). Copper, Cd, Zn, Fe, and Pb concentrations were determined by atomic absorption spectrophotometry (GBS Scientific AVANTA, Dandenong, Australia) using an air/acetylene flame. Each sample was tested in triplicate. High-purity reagents were used in all cases and blanks were analyzed in parallel to validate the efficiency of the method (*Barrera-García et al., 2012*). A quality control sample was analyzed at an interval of every ten samples to ensure the quality of analyses in addition to blanks and calibration standard solutions. Certificate reference material (DORM-2, National Research Council of Canada) was also analyzed in each ten samples run as a quality control (*Jara-Marini, Soto-Jiménez & Páez-Osuna, 2009*); a 93%–106% recovery rate was calculated. The detection limits and quantification limits ($\mu g\,g^{-1}$) were as follows, respectively: Cd: 0.01 and 0.02; Pb: 0.07 and 0.10; Cu: 0.017 and 0.020; Zn: 0.021 and 0.060; Fe: 0.65 and 1.35. All the results are expressed in dry weight.

## Nutritional and risk assessment

The estimated daily intake (EDI, mg trace element $kg^{-1}$ BW $day^{-1}$) of Cu, Cd, Zn and Fe when totoaba muscle is consumed was calculated as follows in Eq. (1) (*Bilandžić et al., 2014*):

$$EDI = (Cm * CR)/BW \qquad (1)$$

where:

Cm = mean concentration of trace metals in the muscle of totoaba, expressed in fresh weight ($\mu g\ g^{-1}$, fw).

CR = Mean per capita daily consumption rate of fish muscle (in this study, fishermen populations living on the coast of north-western Mexico (130 g $day^{-1}$) (*Zamora-Arellano et al., 2018*).

BW = mean body weight in the general population or subpopulation (74 kg; *Zamora-Arellano et al., 2018*).

The contribution of the mean daily consumption of totoaba muscle to the recommended daily intake (RDI) was assessed for the essential elements addressed in this study, including Cu: 0.9 mg $day^{-1}$, Zn: 11 mg $day^{-1}$ and Fe: 8 mg $day^{-1}$ person (*Trumbo et al., 2001*) and RfD (*National Academies of Sciences, Engineering, and Medicine, 2017*).

The hazard index (HI) for Cu, Cd, Zn and Fe based on daily ingestion of muscle of totoaba was achieved by deterministic and probabilistic (Monte Carlo simulations) approaches (*Sanaei et al., 2020*). With the deterministic approach, the contribution of the geometric mean of Cu, Cd, Zn and Fe concentration found in muscle of totoaba was compared to their respective metal reference dose (RfD). The RfD is an estimate (with uncertainty spanning perhaps an order of magnitude) of a daily oral exposure for an acute duration (24 h or less) to the human population (including sensitive subgroups) that is likely to be without an appreciable risk of deleterious effects during a lifetime (*United States Environmental Protection Agency, 2020*). The Environmental Protection Agency (EPA) recommends for Cu 0.40 mg $kg^{-1}day^{-1}$; Cd 0.001 mg $kg^{-1}day^{-1}$; Zn 0.30 mg $kg^{-1}day^{-1}$; and Fe 0.7 mg $kg^{-1}day^{-1}$ (*United States Environmental Protection Agency, 2020*). The EPA has not established an RfD for Pb because a "safe" exposure limit for Pb toxicity is still uncertain (*EFSA, 2010*; *Korkmaz et al., 2019*).

As a probabilistic approach of risk assessment, Monte Carlo simulations (MCS) were performed. MCS recently have been used for environmental health and safety risk assessments associated to metal exposure through water and food (*Jia et al., 2018*; *Chen et al., 2019*; *Sanaei et al., 2020*). MCS reduce the probability of uncertainty caused when only a single point *value* is used for each variable during the risk assessment (as was calculated deterministically using Eq. (1)). The distributions of parameters for MCS are listed in Table 1. MCS mainly consist of three steps, (1) introduction and definition of the distribution functions of the variables; (2) inclusion of the formula that will be used to calculate the risk considering also the RfD, and (3) carrying out several simulations generating random numbers of the unknown parameter from a specific probability density function. Each random number simulates a scenario independently. The process is iterative and the results are displayed as a probability curve, where the X axis represents the simulated

**Table 1  Monte Carlo parameter distributions and parameters used in the health risk assessment.**

| Parameters | Unit | Description | Distribution | Value | Ref |
|---|---|---|---|---|---|
| EDI | $(mg\,kg^{-1})$ $day^{-1}$ | Estimated Daily Intake through ingestion | – | Equation 1 | *Bilandžić et al. (2014)* |
| Bw | Kg | Body weight adult | Log normal | $74 \pm 10$ kg | *Zamora-Arellano et al. (2018)* |
| CR | Kg $day^{-1}$ | Consumption Rate | Log normal | $0.100 \pm 0.030$ | *Zamora-Arellano et al. (2018)* |
| HI | | Hazard Index | – | | *Sanaei et al. (2020)* |
| $C_{metal}$ | mg $kg^{-1}$ | | Log normal | | |
| RfD | $(mg\,kg^{-1})$ $day^{-1}$ | Reference dose of heavy metals | Fixed values | Varied depending on the specific metal | *USEPA (2020)* and *Chen et al. (2019)* |

**Notes.**

EDI, estimated daily intake; Bw, body weight; CR, consumption rate; HI, hazard index; $C_{metal}$, metal concentration in muscle of totoaba; RfD, reference dosis.

variable and the Y axis represents the probability of occurrence (*Chen et al., 2019*). In this study, Cristal Ball (11.1.2.4) software from Oracle was used to perform MCS. Iterations for every run were set to 10,000 (*Jia et al., 2018*; *Sanaei et al., 2020*; *Chen et al., 2019*) and Log-normal probability distribution functions for the parameters (Table 1) were employed to generate the cumulative hazard index (*Chen et al., 2019*) of Cu, Zn, Cd and Fe associated to the consumption of muscle of totoaba.

Based on the results of the MCS, a sensitivity analysis was conducted by introducing variation in some parameters and their effect over hazard index. The level of sensitivity is shown as contribution to variance (%) and correlation. This analysis is employed to judge the significance of the input parameters to the risk estimation (*Chen et al., 2019*; *Sanaei et al., 2020*).

Aiming to identify potential factors that can influence the concentration of Cu, Cd, Zn and Fe in liver and muscle, and therefore in the health risk for consumption of totoaba muscle, a principal component analysis (PCA) was performed using TIBCO STATISTICA software ver. 13.3.

## Statistical analysis

The Shapiro–Wilk's normality and Bartlett's homoscedasticity tests were run. Differences between groups (IF, MF, IM, MM, UN) were examined by means of analysis of variance (ANOVA) followed by *post-hoc* Tukey's honest significance test (*Zar, 1998*). Differences were considered significant when $p < 0.05$. Statistica 8.0 (StatSoft Inc. Tulsa, OK, USA) software was used to perform all statistical analyses. Results are expressed as mean $\pm$ SE.

Generalized linear models (GLM) were used to explore the variables that may affect muscle and liver Cd levels in the totoaba (dependent variables, Table 2) through simultaneous effects of the independent variables (season of the year, sex/maturity stage, total length, and concentrations of trace elements in liver and muscle of totoaba) (Tables 2 and 3). Two models were fitted to the data; the variables selected as continuous dependent (response) variable in each model were liver Cd concentration in model 1, and muscle Cd in model 2. In both cases, the dependent variables were modeled assuming a gamma distribution error since no significant differences were obtained from the Chi square test

**Table 2** Factors hypothesized to affect cadmium (Cd) in liver and muscle of totoaba (*Totoaba macdonaldi*) sampled in the Gulf of California, Mexico.

| Variable | Type | Description |
|---|---|---|
| Season | Categorical | Autumn |
| | | Winter |
| | | Spring |
| Sex/maturity stage | Categorical | MF; mature females |
| | | MM; mature males |
| | | IF; immature females |
| | | IM; immature males |
| Length | Continuous | Length of the organism (g) |
| Cadmium in liver and muscle | Continuous | Metal concentration ($\mu g \cdot g^{-1}$) |
| Copper in liver and muscle | Continuous | Metal concentration ($\mu g \cdot g^{-1}$) |
| Zinc in liver and muscle | Continuous | Metal concentration ($\mu g \cdot g^{-1}$) |
| Iron in liver and muscle | Continuous | Metal concentration ($\mu g \cdot g^{-1}$) |

($p > 0.05$). In the GLM, the season and sex/maturity stage were included as categorical explanatory variables; length and metal concentrations in liver and muscle were included as continuous explanatory variables (Tables 2 and 3). In all cases, the log-type link function was used to relate the dependent variable (liver Cd, muscle Cd) with categorical and continuous independent variables. Finally, the models were visually validated through the residual deviation of observed and predicted values selected based on the proportion of the explained deviance (*Murillo-Cisneros et al., 2018*), and Akaike's and Bayesian information criteria (AIC and BIC, respectively) (*Burnham & Anderson, 2004*; *Hobbs & Hilborn, 2006*; *Johnson & Omland, 2004*; *Zuur et al., 2009*).

## RESULTS

Table 3 shows trace elements concentrations in muscle and liver of totoaba, grouped by habitat (estuaries, rocky and continental), season (winter, spring and autumn), and sex/maturity stage. In liver, Cu levels were significantly higher in MF in winter in estuarine habitat than IF, IM, and UN individuals collected in autumn in continental habitats ($p < 0.05$). Female individuals (mature and immature) in spring had about 3 times lower liver Cd concentrations compared to IF, IM, and UN specimens collected in autumn in rocky habitat; there were no significant differences between the individuals collected in winter and the totoabas collected in spring (both estuarine and rocky habitats) and autumn (continental habitats) ($p > 0.05$). Female totoabas collected in winter showed significantly higher liver Zn levels in comparison with their counterparts collected in spring ($p < 0.05$). Mature females in spring from rocky habitat had significantly lower liver Fe as compared to IF and UN collected in autumn from the same habitat ($p < 0.05$) (Table 3). Also, in autumn, significantly higher concentrations of Fe were observed in organisms from rocky *versus* continental habitats ($p < 0.05$). No detectable Pb levels (<0.07 $\mu g\,g^{-1}$) were observed in any of the analyzed liver samples.

**Table 3** Trace elements concentration ($\mu g g^{-1}$) in liver and muscle of *Totoaba macdonaldi* by season and sex/maturity stage groups.

| Tissue | Habitat | n | Season | Sex/maturity Stage | Copper | Cadmium | Zinc | Iron | Lead |
|---|---|---|---|---|---|---|---|---|---|
| **Liver** | | | | | | | | | |
| | | | Winter | | | | | | |
| | E | 15 | | MF | 44.04 ± 5.02[a] | 1.82 ± 0.30[a] | 116.13 ± 6.26[a] | 1,284 ± 103[abcd] | <0.07 |
| | E | 18 | | MM | 35.84 ± 2.39[ab] | 3.18 ± 1.13[abc] | 107.98 ± 4.14[ab] | 1,187 ± 151[abcd] | <0.07 |
| | | | Spring | | | | | | |
| | E | 8 | | MF | 36.02 ± 8.05[ab] | 0.89 ± 0.38[a] | 80.68 ± 6.81[ab] | 623 ± 137[ab] | <0.07 |
| | E | 3 | | MM | 22.78 ± 5.83[ab] | 0.89 ± 0.73[ab] | 72.26 ± 6.58[ab] | 628 ± 46[abcd] | <0.07 |
| | R | 7 | | MF | 24.23 ± 2.20[ab] | 1.64 ± 0.89[ab] | 69.41 ± 1.52[b] | 556 ± 93[a] | <0.07 |
| | R | 8 | | MM | 20.07 ± 4.43[b] | 1.24 ± 0.37[ab] | 79.86 ± 6.29[ab] | 813 ± 173[abd] | <0.07 |
| | R | 11 | | IF | 21.80 ± 3.34[b] | 0.47 ± 0.36[a] | 86.66 ± 5.95[ab] | 560 ± 59[ab] | <0.07 |
| | R | 6 | | IM | 23.40 ± 6.21[ab] | 1.06 ± 0.70[ab] | 76.55 ± 11.15[ab] | 787 ± 283[abc] | <0.07 |
| | | | Autumn | | | | | | |
| | R | 12 | | IF | 28.87 ± 4.02[ab] | 5.53 ± 0.96[c] | 93.89 ± 11.70[ab] | 1,521 ± 189[bcd] | <0.07 |
| | R | 11 | | IM | 31.34 ± 4.32[ab] | 6.7 ± 1.01[c] | 97.82 ± 10.29[ab] | 1961 ± 336[d] | <0.07 |
| | R | 21 | | UN | 25.89 ± 2.96[b] | 5.33 ± 0.61[c] | 96.00 ± 5.34[ab] | 1,792 ± 181[cd] | <0.07 |
| | C | 9 | | IF | 16.30 ± 2.93[b] | 1.32 ± 0.22[a] | 89.58 ± 8.37[ab] | 400 ± 63[a] | <0.07 |
| | C | 7 | | IM | 11.80 ± 2.64[b] | 1.06 ± 0.09[ab] | 75.74 ± 4.26[ab] | 359 ± 49[a] | <0.07 |
| | C | 6 | | UN | 20.90 ± 4.58[b] | 2.38 ± 0.50[abc] | 113.62 ± 12.84[ab] | 470 ± 92[ab] | <0.07 |
| **Muscle** | | | | | | | | | |
| | | | Winter | | | | | | |
| | E | 14 | | MF | 1.80 ± 0.41 | <0.01 | 17.74 ± 1.00[A] | 22.99 ± 2.84[ABC] | <0.07 |
| | E | 18 | | MM | 1.54 ± 0.32 | <0.01 | 17.78 ± 0.72[A] | 20.43 ± 4.00[ABC] | <0.07 |
| | | | Spring | | | | | | |
| | E | 8 | | MF | 0.86 ± 0.43 | <0.01 | 15.15 ± 0.61[AB] | 21.67 ± 1.95[ABC] | <0.07 |
| | E | 3 | | MM | 1.01 ± 0.54 | <0.01 | 15.73 ± 1.78[AB] | 22.49 ± 3.63[ABC] | <0.07 |
| | R | 7 | | MF | 2.00 ± 0.80 | 0.03 ± 0.02 | 13.86 ± 3.20[AB] | 21.90 ± 5.88[ABC] | <0.07 |
| | R | 8 | | MM | 0.39 ± 0.19 | <0.01 | 11.79 ± 1.16[B] | 12.74 ± 1.55[ABC] | <0.07 |
| | R | 12 | | IF | <0.017 | <0.01 | 12.42 ± 0.42[B] | 11.73 ± 2.13[AB] | <0.07 |
| | R | 6 | | IM | 0.32 ± 0.21 | <0.01 | 13.84 ± 0.49[AB] | 21.90 ± 4.47[ABC] | <0.07 |
| | | | Autumn | | | | | | |
| | R | 12 | | IF | 0.48 ± 0.19 | <0.01 | 12.23 ± 1.44[B] | 7.96 ± 2.73[B] | <0.07 |
| | R | 11 | | IM | 1.68 ± 1.06 | <0.01 | 11.91 ± 1.04[B] | 15.03 ± 4.79[ABC] | <0.07 |
| | R | 21 | | UN | 0.04 ± 0.02 | <0.01 | 11.74 ± 0.59[B] | 10.84 ± 2.54[AB] | <0.07 |
| | C | 9 | | IF | 1.37 ± 0.62 | <0.01 | 11.35 ± 0.65[B] | 18.18 ± 1.58[ABC] | <0.07 |
| | C | 8 | | IM | 0.48 ± 0.35 | <0.01 | 10.95 ± 0.50[B] | 33.17 ± 6.50[C] | <0.07 |
| | C | 7 | | UN | 0.69 ± 0.46 | 0.15 ± 0.14 | 13.07 ± 0.66[AB] | 31.19 ± 5.36[AC] | <0.07 |

**Notes.**

Totoaba groups: MF, mature females; MM, mature males; IF, immature females; IM, immature males; UN, undifferentiated. Different letters indicate significant differences ($p < 0.05$); lowercase letters denote differences within a habitat and season between maturity stages for each sex in liver; uppercase letters denote differences within a habitat and season between maturity stages for each sex in muscle. E, estuarine; R, rocky; C, continental. Values are shown as means ± standard error.

In totoaba muscle, Pb levels were below the detection limit ($<0.07 \ \mu g \ g^{-1}$), and only 10% of the samples showed detectable Cd levels ($>0.01 \ \mu g \ g^{-1}$) (Table 3). No significant differences were observed in muscle Cu and Cd levels ($p > 0.05$). Mature males and MF had significantly higher muscle Zn levels in winter as compared to organisms collected both in spring and autumn in rocky and continental habitats ($p < 0.05$).

Copper, Zn, and Fe concentrations were significantly higher in liver than in muscle ($p = 0.001$) (Table 3). While 100% of liver samples showed detectable Cu and Cd levels, these elements were undetectable in muscle in 40% and 90% of samples, respectively. Liver Cu, Zn, and Fe concentrations were approximately up to 40-, 7- and 80-fold higher, respectively, than in muscle.

## Health risk

Figure 3 shows the Cu, Cd, Zn and Fe content in muscle of *Totoaba macdonaldi* relative to the RDI, by sex/maturity stage groups and season (winter, spring, autumn). Consumption of totoaba muscle can provide the human diet with approximately 70% of Cu, 60% of Zn, and more than 100% of Fe of the RDI. In addition, the concentrations of Cu, Zn and Fe in muscle of totoaba are from one to several orders of magnitude below the RfD. Cadmium input from totoaba muscle to the human diet can be from zero to approximately 20% of the RfD, depending on the fish's sex/maturity stage. No detectable concentrations of Pb were recorded in muscle of totoaba; therefore, no comparisons to RfD are available.

The potential health risk to local inhabitants from long-term exposure for Cu, Cd, Zn and Fe in fish muscles of totoaba was assessed deterministically (*Sanaei et al., 2020*) and through the hazard index (HI) obtained by the probabilistic approach using MCS in adult population. Monte-Carlo simulation was introduced to conduct uncertainty through 10,000 iterations for each trace element simulation. The probabilistic distributions of health risk as hazard index (HI) obtained from MCS are demonstrated in Fig. 4, including mean, 5th percentile and 95th percentile for each metal. The HI values obtained from probabilistic values (MCS) are lower than those calculated by deterministic approach; Cu: 0.08 vs 0.10; Cd: 0.06 vs 0.07; Zinc: 0.17 vs 0.21 and Fe 0.10 vs 0.13, respectively. The percentile values for the HI for all the metals are lower than 1. In Fig. 5 the associated risk given by the HI of Cu, Cd, Zn and Fe from totoaba muscle consumption is shown. The contribution of each HI can be arranged in decreasing order as follows $HI_{Zn} > HI_{Fe} > HI_{Cu} > HI_{Cd}$. The sensitivity analysis (Table 4) showed that the consumption rate of muscle of totoaba is the parameter that contributes with the highest explanation of the variance of each metal HI, 75.4% in $HI_{Cu}$; 82.7% in $HI_{Cd}$, 82.7 en $HI_{Zn}$ and 83% in $HI_{Fe}$. Otherwise, the body index has a negative relationship and in the case of totoaba muscle the influence of metal content has the lowest importance in the contribution of the variance.

For qualitative evaluation of clustering behavior of Cu, Cd, Zn and Fe in liver and muscle of totoaba regarding season, habitat and gonadic state a principal component analysis (PCA) when varimax normalization was applied. The results are shown in Table 5. The PCA of these data indicates their association and grouping in three factors that explain 52.28% of the variance. Factor 1 contributed 24.60% of the total variance with the highest loadings of $Cu_{muscle}$, $Cu_{liver}$, and $Zn_{muscle}$, that are directly related to winter, estuarine,

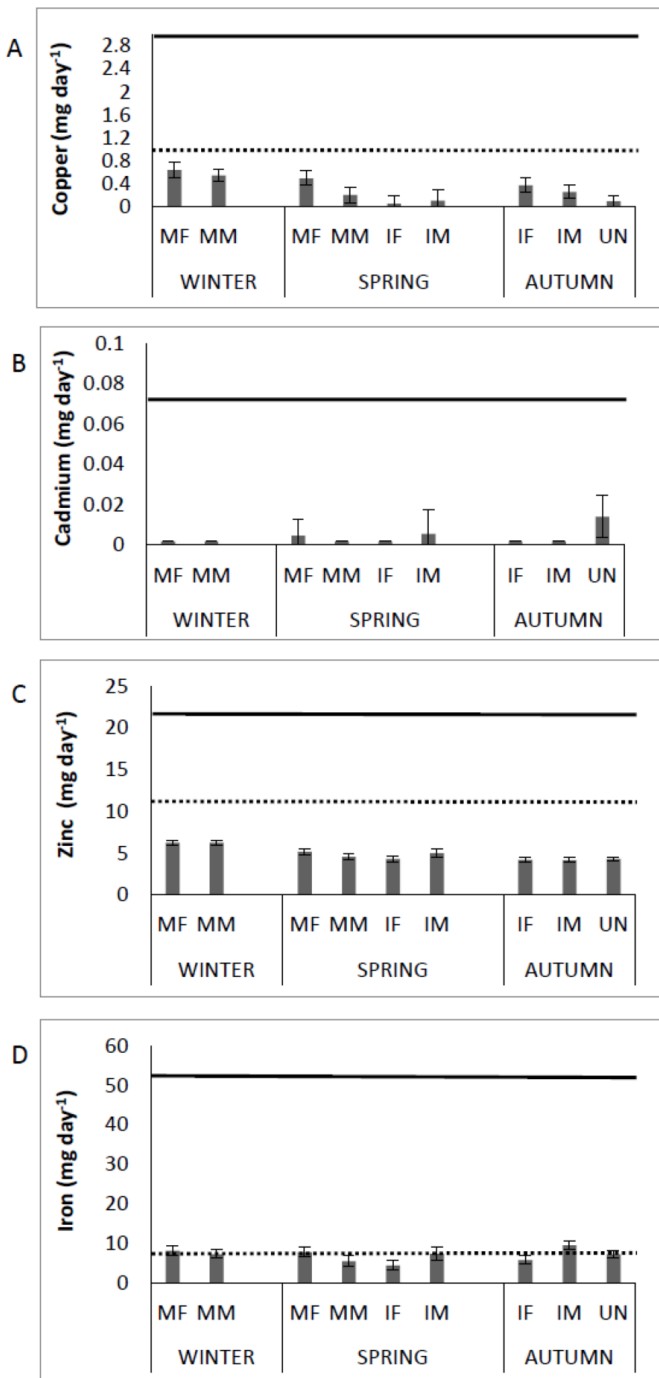

**Figure 3** Contribution of copper, cadmium, zinc and iron by consumption of muscle of *Totoaba macdonaldi* ($\mu$g day$^{-1}$) to the dietary reference intakes (DRI, *National Academies of Sciences, Engineering, and Medicine, 2017*; dotted line) and reference dose. (RfD, *USEPA, 2020*; continuous line) by sex/maturity stage groups and season (spring, autumn and winter). IF, immature female; MF, mature female; IM, immature male; MM, mature male; U, undifferentiated.

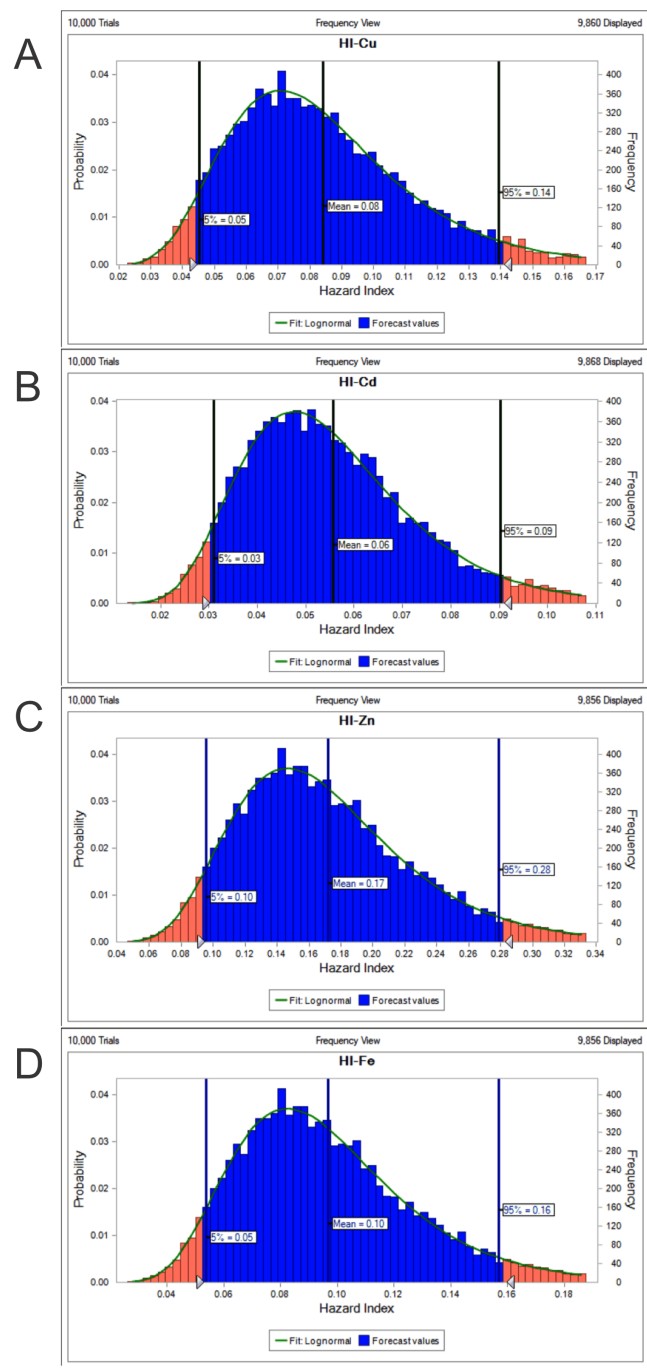

**Figure 4** Histograms of the cumulative distribution of the 5%, mean and 95% hazard indexes (HI) of copper (Cu), cadmium (Cd), zinc (Zn) and iron (Fe) for consumption of muscle of *Totoaba macdonaldi*.

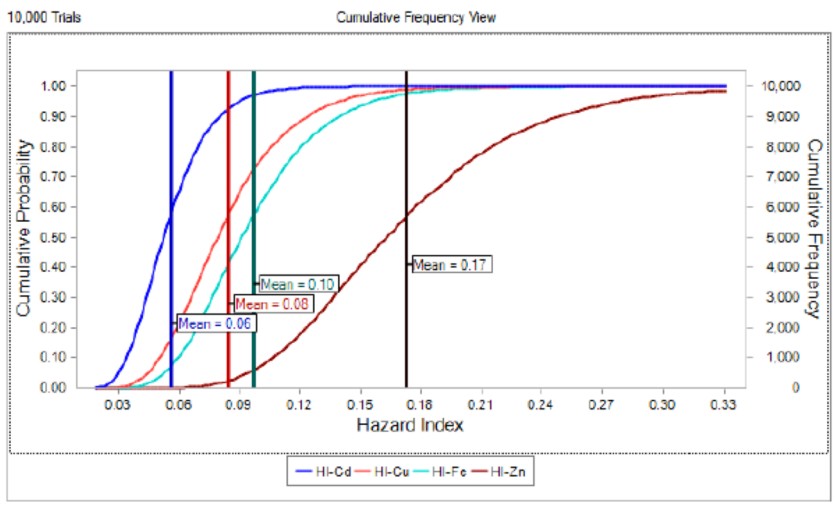

**Figure 5** Cumulative probability distribution curve of hazard index for copper (Cu), cadmium (Cd), zinc (Zn) and iron (Fe) for consumption of *Totoaba macdonaldi* muscle.

**Table 4 Sensitivity analyses of input parameters for the effect in the hazard index (HI) as estimation of the health risk for consumption of muscle of totoaba.** The level of sensitivity is shown as contribution to the variance (%) and rank correlation (10,000 trials; Crystal Ball).

|  | Parameters | Contribution to Variance (%) | Rank correlation |
|---|---|---|---|
| **Sensitivity: HI$_{copper}$** | Consumption Rate | 7.54E−01 | 8.53E−01 |
|  | Body weight | 1.60E−01 | −3.92E−01 |
|  | Copper$_{muscle}$ | 8.56E−02 | 2.87E−01 |
| **Sensitivity: HI$_{cadmium}$** | Consumption Rate | 8.27E−01 | 8.98E−01 |
|  | Body weight | 1.72E−01 | −4.10E−01 |
|  | Cadmium$_{muscle}$ | 1.34E−05 | 3.62E−03 |
| **Sensitivity: HI$_{zinc}$** | Consumption Rate | 8.27E−01 | 8.98E−01 |
|  | Body weight | 1.72E−01 | −4.10E−01 |
|  | Zinc$_{muscle}$ | 1.74E−05 | 4.12E−03 |
| **Sensitivity: HI$_{iron}$** | Consumption Rate | 8.30E−01 | 9.01E−01 |
|  | Body weight | 1.70E−01 | −4.07E−01 |
|  | Iron$_{muscle}$ | 6.95E−05 | 8.24E−03 |

mature females and mature male, but inversely related to spring, and in less magnitude, to immature organisms. Factor 2 contributed 15.88% of the total variance with the highest loading of Cd$_{liver}$, Zn$_{liver}$, and Fe$_{liver}$ that are only associated to autumn and to a lesser extent spring. Factor 3 has the highest loading of Fe$_{muscle}$ that is inversely to autumn and to rocky habitat but directly related to continental habitat.

**Table 5   Loadings for season (winter, spring and autumn), habitat (estuarine, rocky, continental), sex/-maturity stage (male, female, mature, immature, undifferentiated) and estimated dairy intake (EDI) of trace metal content in muscle in totoaba on VARIMAX-rotated factors by principal components analysis.** The variables that contribute the most to each factor are marked with an asterisk.

| Variable | Factor 1 | Factor 2 | Factor 3 |
| --- | --- | --- | --- |
| Winter | −0.864* | 0.201 | 0.114 |
| Spring | 0.726* | 0.496 | 0.380 |
| Autumn | 0.001 | −0.724* | −0.518 |
| Estuarine | −0.932* | −0.002 | 0.049 |
| Rocky | 0.632 | 0.074 | −0.664* |
| Continental | 0.316 | −0.099 | 0.848* |
| Mature females | −0.541* | −0.219 | −0.095 |
| Mature male | −0.505* | −0.084 | −0.112 |
| Immature females | 0.389* | −0.146 | −0.040 |
| Immature males | 0.294* | 0.006 | 0.146 |
| Undifferentiated | 0.401 | 0.445* | 0.144 |
| $Copper_{muscle}$ | −0.332* | −0.111 | 0.225 |
| $Cadmium_{muscle}$ | 0.069 | 0.018 | 0.251* |
| $Zinc_{muscle}$ | −0.700* | 0.100 | 0.070 |
| $Iron_{muscle}$ | −0.281 | −0.206 | 0.524* |
| $Copper_{liver}$ | −0.511* | 0.434 | −0.248 |
| $Cadmium_{liver}$ | 0.149 | 0.788* | −0.224 |
| $Zinc_{liver}$ | −0.322 | 0.663* | −0.006 |
| $Iron_{liver}$ | −0.011 | 0.780* | −0.318 |
| Eigenvalue | 4.67 | 3.02 | 2.24 |
| Explained variance (%) | 24.60 | 15.88 | 11.81 |
| Cumulative variance | 24.60 | 40.48 | 52.28 |

## Generalized linear models (GLM)

Due to Cd being an important element in relation to health risk, GLM were used to explore the variables that may affect the content of Cd in muscle and liver in totoaba. The first GLM model explains 46% of the variance (Table 6), fitting the data reasonably well. The ratios of the deviance and Pearson Chi square over the degree of freedom close to 1.0, in particular model 2, does not show evidence of overdispersion. The model suggests that liver Cd concentration varies inversely with liver Zn concentration. Other elements measured in liver or muscle samples were not identified as significantly affecting liver Cd content. However, the winter season and sex/maturity stage and two interactions were significant. The second GLM model (Table 6), which explains 71% of the variance, suggests that liver Cd is the only element that significantly affects muscle Cd concentrations in totoaba. Furthermore, sex/maturity stages and an interaction were significant. Both models were validated using the analysis of residuals (Fig. 6), suggesting that the variance is homogeneous for all the independent variables and that the relationship between the observed and fitted values is nearly linear.

**Table 6  Generalized linear models explaining the variability of cadmium (Cd) in liver and muscle of *Totoaba macdonaldi*.** Values marked with * were significantly different ($p < 0.05$) from the intercept.

| | Models | |
|---|---|---|
| **Statistics** | **1; Cd in liver** | **2; Cd in muscle** |
| Error | Gamma | Gamma |
| Link | Log | Log |
| Deviance | 172 | 73 |
| Explained deviance | 46% | 71% |
| Pearson Chi$^2$ | 103 | 76 |
| Log likelihood | −167 | 505 |
| Df | 100 | 74 |
| Deviance/Df | 1.72 | 0.98 |
| Pearson's Chi$^2$/Df | 1.03 | 1.03 |

| **Level of Effect** | **Estimate** | **s. e.** | **Estimate** | **s. e.** |
|---|---|---|---|---|
| Intercept | −4.04* | 0.92 | −5.93* | 0.73 |
| Copper in muscle | −0.06 | 0.08 | −0.07 | 0.06 |
| Cadmium in muscle | 0.70 | 1.18 | – | – |
| Zinc in muscle | −0.02 | 0.03 | −0.04 | 0.03 |
| Iron in muscle | 0.004 | 0.01 | 0.01 | 0.008 |
| Length | 0.00006* | 0.00002 | 0.00002 | 0.46 |
| Copper in liver | 0.02 | 0.009 | −0.008 | 0.007 |
| Cadmium in liver | – | – | 0.14* | 0.05 |
| Zinc in liver | 0.01* | 0.006 | −0.001 | 0.005 |
| Iron in liver | 0.0003 | 0.0002 | 0.0001 | 0.0002 |
| Season 1 (winter) | −2.98* | 0.58 | 0.33 | 0.40 |
| Season 2 (spring) | 0.31 | 0.45 | 0.14 | 0.24 |
| Sex/maturity stage (MF) | 1.28* | 0.49 | 2.16* | 0.44 |
| Sex/maturity stage (MM) | 1.79* | 0.41 | 1.08* | 0.37 |
| Sex/maturity stage (IM) | 0.45 | 0.70 | 1.35* | 0.45 |
| Sex/maturity stage (IF) | 0.83* | 0.39 | −1.18* | 0.27 |
| Season 1*Sex/maturity stage | 3.50* | 0.95 | 2.21* | 0.58 |
| Season 2*Sex/maturity stage | 1.88* | 0.96 | – | – |
| AIC[a] | | 386 | | −964 |
| BIC[b] | | 459 | | −905 |

# DISCUSSION

Compared to muscle, liver accumulates a larger concentration of metals because of its detoxifying function and specific metabolic rate, acting as a depot of chemical elements, some of which function as enzyme cofactors (*Kalay, Ay & Canli, 1999*). This held true for totoaba analyzed in the present study; liver concentrations of Cu, Zn and Fe were 40-, 7-, and 80-fold higher than in muscle. The liver is highly active in fish metabolism and, therefore, the liver may accumulate metals to higher levels than other tissues, such as the muscle (*Kalay & Canli, 2000*).

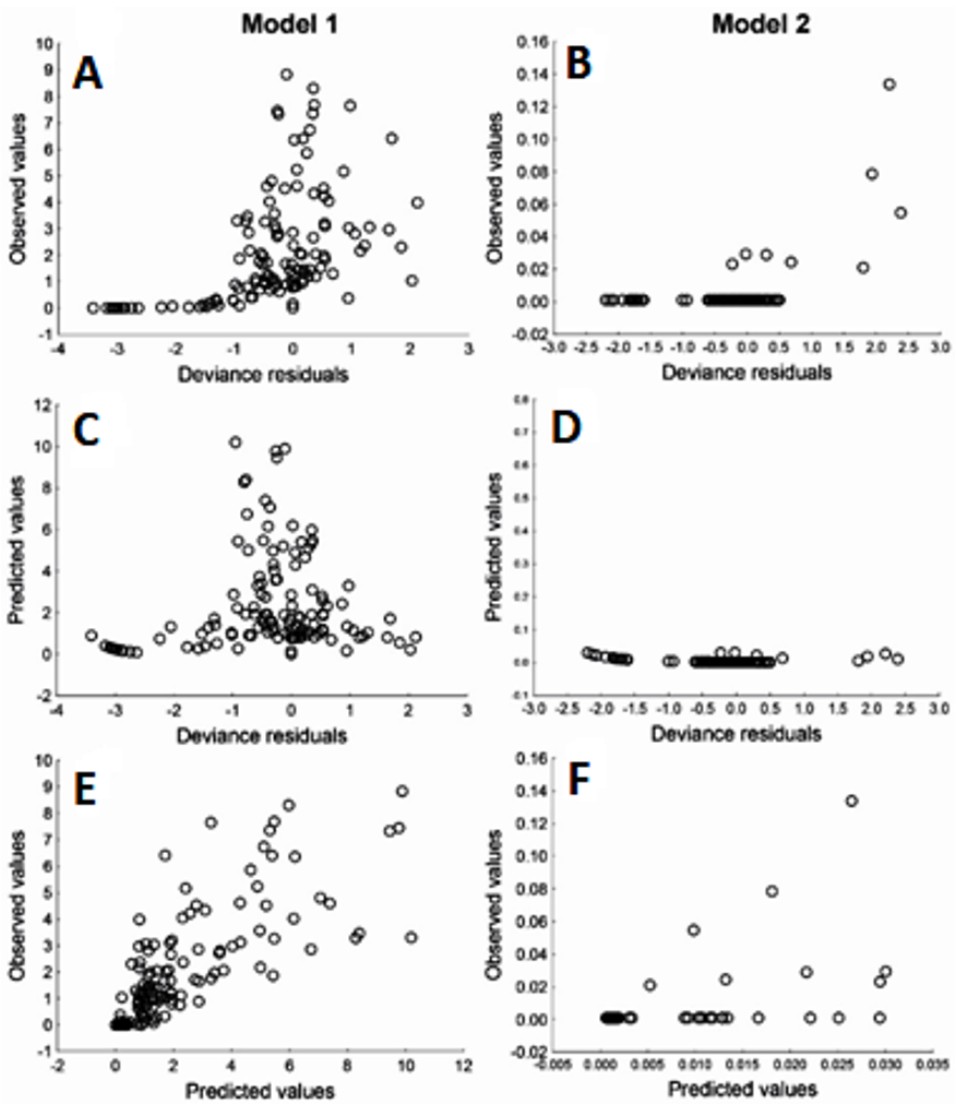

**Figure 6** Residuals analysis of the generalized linear models, assuming a gamma error distribution, generated for cadmium (Cd) in liver (model 1: A, C, E) and muscle (model 2: B, D, F) of *Totoaba macdonaldi* in the Gulf of California, Mexico.

In this study, no detectable Pb levels were recorded in either liver or muscle of totoaba. Other scianid fish, such as *Otolithes ruber* and *Johnius belangerii,* also show a low Pb content relative to other fish species, including *Pampus argenteus* (Stromateidae), *Pomadasys sp.* (Haemulidae), *Euryglossa orientalis* (Soleidae), and *Cynoglossus arel* (Cynoglossidae) collected in the same sites from the Persian Gulf (*Agah et al., 2009*; *Monikh et al., 2012*). In biological systems, trace elements may be either bound to molecules or found as chemical fractions (speciation) that cannot be assimilated by the organism; in which case, trace elements are not bioavailable, can be excreted, and are not bioaccumulated (*Bebianno & Langston, 1991*). Although Pb has been reported at high concentrations in invertebrates

such as clams (up to $9.2 \pm 1.4$ µg g$^{-1}$ dry weight) from the Upper Gulf of California (*Cadena-Cárdenas et al., 2009*), in fishes of this region the levels are characteristic of areas not contaminated by this element (*Ruelas-Inzunza & Páez-Osuna, 2008*; *Jara-Marini, Soto-Jiménez & Páez-Osuna, 2009*; *Ruelas-Inzunza, Páez-Osuna & García-Flores, 2010*). Apparently, Pb does not undergo biomagnification across some food chains in the Gulf of California (*Jara-Marini, Soto-Jiménez & Páez-Osuna, 2009*). This may be due to the functional (carboxyl, sulfhydryl) groups to which Pb can bind in mollusk and fish species (*Ruelas-Inzunza & Páez-Osuna, 2008*) that may be found as part of the totoaba diet (*De Anda-Montañez et al., 2013*).

The highest Cu levels in liver and muscle of totoaba quantified in this study were below 70 µg g$^{-1}$. Natural Cu deposits are found along the Baja California peninsula, particularly in Santa Rosalia, Baja California Sur. The Cu concentrations in totoaba in this study were found to be similar to those recorded in other fish, such as *O. ruber* (7.5–19.5 µg g$^{-1}$), *J. belangerii* (0545–1458 µg g$^{-1}$), *Menticirrhus americanus* (8.8–18 µg g$^{-1}$), *Micropogonias furneiri* (10–122 µg g$^{-1}$), and *Cynoscion guatucupa* (22–119 µg g$^{-1}$) (*Monikh et al., 2012*; *Agah et al., 2009*; *Viana, Huertas & Danulat, 2005*). These Cu concentrations are considered suitable for human consumption (approximately 120 µg g$^{-1}$ dry weight; 30 µg g$^{-1}$ ww, (*Viana, Huertas & Danulat, 2005*; *Monikh et al., 2012*) even though in these studies, fish were sampled in areas influenced by wastewater from tanning, chemical, and agricultural industries, or from urban effluents (*Agah et al., 2009*; *Viana, Huertas & Danulat, 2005*). Variations in tissue Cu levels in sciaenids are attributed to the physiology of each species, rather than to the influence of the environmental conditions in the area where they live (*Hamza-Chaffai et al., 1995*). In totoaba in this study, significantly higher Cu content in MF as compared to IF is likely due to Cu functioning also as a cofactor of cytochrome c oxidase (*Kirchgessner et al., 1977*); the activity of this enzyme increases in the reproductive stage contributing to meet the additional energy requirements during this process in fishes (*Olsson, Haux & Förlin, 1987*).

Mature female and male totoaba had higher Zn concentrations in winter than in spring. In the squirrelfish (*Sargocentron spiniferum*), a reef-associated fish, and the euryhaline rainbow trout (*Oncorhynchus mykiss*), a freshwater fish, liver Zn content bound to metallothionein increases, more so in MF than in MM, during reproduction, potentially aiding vitellogenesis and the synthesis of estradiol-17$\beta$ (*Thompson et al., 2001*). The demand for Zn increases during embryonic and neonatal development (*Olsson, Haux & Förlin, 1987*; *Thompson et al., 2002*). In spring, Fe content in the liver of MF and MM was almost 50% the content observed in totoaba collected in winter. This difference in Fe levels could be associated with fasting conditions reported for totoaba in its spawning grounds in the north of the Upper Gulf of California over the winter (*De Anda-Montañez et al., 2013*). The feeding rate of totoaba adults has been reported to decrease during the reproductive season (*Román Rodríguez & Hammann, 1997*).

The Cd concentration in the liver of immature totoaba collected in autumn in rocky habitat was more than three-fold the level in organisms collected in spring ($p < 0.05$). In a study that compared Cd content between *J. belangerii* (Sciaenidae), *Liza abu* (Mugilidae), and *Euryglossa orientalis* (Mugilidae), the sciaenid fish showed significantly higher Cd levels

$(3.80 \pm 0.66\ \mu\mathrm{g\,g^{-1}})$, which was attributed to its feeding habits, because *J. belangerii* lives in close association with sediment, and feeds mainly upon crustaceans, mollusks, and shrimp (*Monikh et al., 2012*). In autumn, totoaba feeds in the area adjacent to the large islands; the oceanographic conditions in this area foster a high primary productivity, reflected as a high abundance of birds and marine mammals (*Hidalgo-González & Alvarez-Borrego, 2004*). This photosynthetic activity is maintained in part by nutrient enrichment, including Cd (*Álvarez Molina, 2013*; *Delgadillo-Hinojosa et al., 2001*). In this area, totoaba feed mainly on anchoveta *Cetengraulis mysticetus* and silverside *Colpichthys* spp (*De Anda-Montañez et al., 2013*), which in turn feed on plankton known to contain around $17\ \mu\mathrm{g\,g^{-1}}$Cd (*Lane et al., 2005*). The Cd assimilated by fish accumulates in the liver; in which it is bound mainly to metallothionein, a protein of low molecular weight with a high content of amino and sulfhydryl groups (*Hamza-Chaffai et al., 1995*). These proteins play a central role in the regulation and detoxification of trace elements. Divalent cations such as Cd, have high affinity for sulfhydryl groups in metallothionein; as a result, Cd displaces essential elements such as Zn, which upon release boost the synthesis of additional metallothionein (*Bebianno & Langston, 1991*). The factors obtained by the PCA analysis showed that trace elements could be associated mainly to the reproductive physiology of the organisms as well as the starving/feeding conditions during the different seasons and in the different habitats. The reproduction process demands some metals that are required as co-factors of several enzymatic reactions involved in metabolic pathways (*Olsson, Haux & Förlin, 1987*; *Farkas, Salanki & Specziar, 2002*; *Canli & Atli, 2003*). The combined results from this study suggest that the observed variations in Cd, Cu, Zn and Fe concentrations in totoaba liver are related to the feeding and reproductive physiology of this species rather than to environmental exposure.

In this study, Cu, Cd, Zn, Fe, and Pb contents in totoaba muscle were within the ranges reported in fish from unpolluted areas of the Gulf of California (*Ruelas-Inzunza, Páez-Osuna & García-Flores, 2010*). No significant variations in Cu and Cd content were observed in muscle neither significant loadings were obtained in the PCA indicating that the content of these elements in muscle is unaffected by season (potentially, migration) or by the maturity stage of the totoaba (*Agah et al., 2009*; *Ruelas-Inzunza, Páez-Osuna & García-Flores, 2010*), In this study, in the factor 1 in the PCA, the content of Zn in muscle is associated especially with mature organisms in winter. Zinc is an essential component of metalloenzymes involved in the energy metabolism that contributes with muscle strength and resistance to fatigue (*Krotkiewski et al., 1982*).

In the PCA, in factor 2 the highest loadings of $Cd_{liver}$, $Zn_{liver}$ and $Fe_{liver}$ that can be associated to the nutrition status of the totoaba, due to their starving/feeding periods, are also influenced by changes in their diet composition throughout the life cycle of these organisms (*Farkas, Salanki & Specziar, 2002*). Factor 3 could be associated with environmental conditions in the rocky and continental habitats that influence the Fe content in muscle of totoaba. All the HI obtained by MCS were HI <1; this suggests a negligible health effect (*Sanaei et al., 2020*; *United States Environmental Protection Agency, 1989*) of Cu, Cd, Zn, Fe or Pb content in muscle of totoaba consumed in human diet. The sensitivity analysis agrees with several previous studies reporting that the ingestion rate,

rather than the concentration of the element in food, is more sensitive; therefore, small changes in the consumption rate would affect HI to a greater extent (*Sanaei et al., 2020*; *Chen et al., 2019*; *Guo, Zhang & Wang, 2019*).

Muscle Cu, Cd, Zn, Fe, and Pb concentrations were below the maximum permissible levels for human consumption established by the World Health Organization (WHO) and the European Community for seafood; i.e., 10 $\mu$g g$^{-1}$ww for Cu (Australian legislation; *Nauen, 1983*), 50 $\mu$g g$^{-1}$ ww for Zn (*Collings, Johnson & Leah, 1996*), 0.05 and 0.2 $\mu$g g$^{-1}$ ww for Cd and Pb, respectively (European Community, *EC, 2005*; *WHO, 1996*). Thus, consumption of totoaba does not pose a public health risk. Furthermore, depending on the sex/maturity stage of totoaba, consumption of this fish's muscle may provide approximately 70% Cu, 60% Zn and 100% Fe of the recommended DRI. Both RDA and RfD are comparable to the values obtained in muscle of several fish species from the Mediterranean Sea and the Gulf de California (*Bilandžić et al., 2014*; *Korkmaz et al., 2019*; *Zamora-Arellano et al., 2018*). It is calculated that 130 g of totoaba muscle can provide approximately 20% of the daily tolerable dose of Cd.

In a previous study, liver superoxide dismutase (SOD) activity was found to be related to totoaba's maturity stage (*Hernández-Aguilar et al., 2018*); thus, it was used in this study to further discriminate between environmental and physiologic effects on trace element content in totoaba. This third GLM explains 55% of the variance and suggests that in liver, SOD activity is inversely related to Cd concentration in winter and spring ($p < 0.05$) but not in autumn ($p > 0.05$). The GLMs constructed in this study suggest that liver Cd concentration varies inversely with liver Zn concentration, that liver Cd content affects muscle Cd concentrations, and that in the liver, SOD activity is inversely related to Cd concentration and length of the organisms. The totoaba collected in autumn have the highest liver Cd concentrations, and the highest SOD activity (*Hernández-Aguilar et al., 2018*). In autumn, immature and undifferentiated totoabas feed on sardines and crustaceans (*Cisneros-Mata, Montemayor-Lopez & Roman-Rodriguez, 1995*). In the Upper Gulf of California, in winter and spring, the feeding rate of totoaba is reduced as a result of the reproductive activity (*Román Rodríguez & Hammann, 1997*). The liver is the primary organ where trace elements are stored and physiologically regulated in fish (*Hamza-Chaffai et al., 1995*; *Roméo et al., 1999*); it also participates in antioxidant defense mechanisms (including SOD) that protect tissues from the deleterious effects of free radicals (*Basha & Rani, 2003*; *Halliwell & Gutteridge, 2007*). In the present study, Cd content in totoaba liver may be indicative of the presence of this metal in the environment, but the content of Cu, Zn, and Fe may be associated primarily with the physiology of this fish rather than with the presence of these elements in the environment, as has been reported for other marine fish species (*Hamza-Chaffai et al., 1995*).

## CONCLUSIONS

In summary, although totoaba migrates throughout the Gulf of California, where sites enriched by trace elements are common, Cd, Cu, Zn, Fe and Pb content in liver and muscle of totoaba were within the levels considered as indicators of sites without contamination.

Significantly higher Cu content in mature as compared to immature female totoaba was observed. The highest liver Cd concentrations were recorded at the time that totoaba are feeding around the large islands and the lowest Cd levels in liver were found during the reproductive stage. In muscle, significant variations were observed in Zn content associated with the maturity stage of the female totoaba. Therefore, variations in the content of Cd, Cu, Zn and Fe in liver of totoaba seem to be more related to the feeding and reproductive physiology of this species than as result of environmental exposure; further totoaba feeding ecology studies are suggested to confirm this. Results suggest that consumption of totoaba muscle does not pose a public health risk. Furthermore, depending on the sex/maturity stage of totoaba, this fish's muscle may provide approximately 70% Cu, 60% Zn and 100% Fe of the recommended DRI.

## ACKNOWLEDGEMENTS

Authors acknowledge L. Campos Dávila, J. J. Ramírez Rosas, N. O. Olguín Monroy, H. Bervera León, F. Valenzuela Quiñonez, O. U. Rodríguez García for their invaluable assistance in the field; B. Acosta Vargas and G. Peña Armenta for assistance with lab work; G. Hernández García and A. Landa Blanco for graphic art, and M. E. Sánchez-Salazar for edition of the English manuscript.

### Funding

This study was supported by Consejo Nacional de Ciencia y Tecnología (CONACYT, Grant 2011-01/165376). The funders had no role in study design, data collection and analysis, decision to publish, or preparation of the manuscript.

### Grant Disclosures

The following grant information was disclosed by the authors:
Consejo Nacional de Ciencia y Tecnología: 2011-01/165376.

### Competing Interests

The authors declare there are no competing interests.

### Author Contributions

- Lia C. Méndez-Rodríguez and Juan A. de Anda-Montañez conceived and designed the experiments, performed the experiments, analyzed the data, prepared figures and/or tables, authored or reviewed drafts of the paper, and approved the final draft.
- Berenice Hernández-Aguilar performed the experiments, analyzed the data, prepared figures and/or tables, authored or reviewed drafts of the paper, and approved the final draft.
- Eduardo F. Balart analyzed the data, authored or reviewed drafts of the paper, and approved the final draft.
- Martha J. Román-Rodríguez performed the experiments, analyzed the data, authored or reviewed drafts of the paper, and approved the final draft.

- Tania Zenteno-Savín analyzed the data, prepared figures and/or tables, authored or reviewed drafts of the paper, and approved the final draft.

### Animal Ethics

The following information was supplied relating to ethical approvals (i.e., approving body and any reference numbers):

Mexican government SGPA Dirección General de Vida Silvestre issued scientific collection permits SGPA/DGVS/02913/10, SGPA/DGVS/05508/11 and SGPA/DGVS/00039/13 for this study. These permis were issued based on based on the research protocol described in the project entitled "Health and Conservation status of the totoaba population (Totoaba macdonaldi) in the Gulf of California: a critically endangered species" (CONABIO: FB1508/HK050/10; CONACYT 2011-01/165376; CIBNOR:137C/EP0.04). This study was performed following the demands, requirements and protocols that are required by the various institutions involved (SGPA/DGVS, CONABIO, CONACYT, CIBNOR).

### Field Study Permissions

The following information was supplied relating to field study approvals (i.e., approving body and any reference numbers):

Mexican government SGPA Dirección General de Vida Silvestre issued scientific collection permits SGPA/DGVS/02913/10, SGPA/DGVS/05508/11 and SGPA/DGVS/00039/13 for this study.

### Data Availability

Raw data are available as a Supplemental File.

### Supplemental Information

Supplemental information for this article can be found online at http://dx.doi.org/10.7717/peerj.11068#supplemental-information.

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
