# Peer review of "Influence of sex and maturity state on trace elements content in liver and muscle of the Sciaenidae Totoaba macdonaldi"

_PeerJ, doi:10.7717/peerj.11068_

## Round 0.1 · original submission · Major Revisions

Thank you authors for submitting your manuscript, titled 'Nutritional content of Totoaba macdonaldi (Gilbert, 1890). Part I, Trace elements in liver and muscle ' for consideration at PeerJ.

Reviewers have considered your manuscript. Reviewers' decisions ranged from major revisions to reject. It is clear that a number of issues have been raised. Kindly do a careful and critical evaluation of reviewers' comments, and address concerns raised diligently.
In addition, the editor has the following remarks to add:

1) Given that this is part 1, which addresses trace elements in liver and muscle, kindly make this current work part 2.

2) Kindly look carefully at the following points, and address these carefully:
a) Title: kindly amend the title to: Nutritional content of Totoaba macdonaldi (Gilbert, 1890). Part I, Trace element quantification in liver and muscle. This is because quantification has been performed, using sex maturity, and geo-distribution
b) Introduction: kindly allocate the paragraphs according to this thought-process pathway: i) paragraph 1 can start with introducing Totoaba macdonaldi (Gilbert, 1890) and its relevance, some of which has been addressed already in this paper, first paragraph ending with However, information on the nutritional properties of totoaba is scarce (please delete all that has to with accompanying paper, part 1 should stand alone, from part 2); ii) paragraph 2 can focus on trace elements , some of which has been addressed already in this paper, with more focus on not only how these trace elements come by, but also, why these trace elements come by into the fish product (and that is where Sex and reproductive stage can be included); iii) paragraph 3 can focus on geo-distribution of the fish globally, narrowed down to areas/regions already mentioned , like Gulf of California, where trace element concentrations change due to natural phenomena. That is say that lines 87-93 should be part of the previous paragraph; iv) paragraph 4 should reflect on 'why should changes in the concentrations of essential elements (like Cu, Zn, Fe), and toxic elements for humans (such as Cd and Pb) in fish products be assessed? Which other studies have assessed such, and what was the take-home message ? Why should a study be carried out to assess changes in the concentrations of essential elements (Cu, Zn, Fe) and toxic elements for humans (such as Cd and Pb) in relation to the sex/maturity stage, season and habitat in liver and in the edible tissue (muscle) of totoaba? These should be provided before stating the objective of the study.
Please remove this statement: The content of trace elements in liver and muscle, and of antioxidants and lipid peroxidation in muscle (Conde-Guerrero, et al., accompanying paper) of totoaba was quantified along its migratory route.
c) Materials & Methods: Please start the materials & methods with a sub-section captioned 'overview of experimental program', comprise three sentences, supported by a schematic diagram/ representation showing sample collection> season collection +spawning grounds + non-spawning areas + three types of habitats (please think through this , because this will show exactly how the samples were collected) > Extraction of liver and muscle samples + sex identification and maturity stage determination (also think through this as well)> Trace element analysis (please, identify them exactly based on number of samples, as well as organs sampled) (apply your discretion here...
The purpose of this schematic diagram is to provide the reader exact snapshot of pathway of this research, so to clear all form of ambiguity.
Nutritional risk assessment should be before statistical analysis

Results: Presentation of results should strictly be presentation of results, with no references cited. This sentence, for example, is the discussion: 'The third model explains 55% of the variance and suggests that in liver, SOD activity (Figure 2, data taken from (Hernández-Aguilar et al., 2018) is inversely related to Cd concentration in winter and spring (p < 0.05) but not in autumn (p > 0.05)'.

Discussion: Please distinguish paragraph by paragraph using indentation , for clarity.

Conclusion: Please, put forward recommendation for future studies

Remember this is part 2 (the previous study of antioxidants and lipid peroxidation in muscle is part 1). Look forward to seeing the revised manuscript. This is a brilliant study.

Reviewer 1 ·

Basic reporting

.

Experimental design

.

Validity of the findings

.

Additional comments

Data obtained in this study is worthwhile but not exhaustively interpreted.

Authors should have used Principal Component Analysis to model the risk assessment.

Monte Carlo modeling should also be used.

Include LOD & LOQ and information on QA/QC of data.

Reviewer 2 ·

Basic reporting

No comment

Experimental design

No comment

Validity of the findings

No comment

Additional comments

The reviewer has carefully read through and reviewed the manuscript. Although the manuscript reports the concentrations of several elements in the liver and muscle of fish species, the reviewer could not find the importance and significance of this regional study as a research publication for The PEERJ Journal. Therefore, the manuscript was judged as Rejection.

Reviewer 3 ·

Basic reporting

no comment

Experimental design

no comment

Validity of the findings

good results obtained

Additional comments

General comment:
the results obtained are good and important for society, scientists and industry thus for publication. congratulation.

Introduction
Line 56: antioxidant and lipid peroxidation is not necessary as it will appear in the second paper


Material and Methods
- Line 108: CIBNOR, what is this about and is the permits number available?
- Line 109-110: “...and alway conducting...endangered species.” is not necessarily needed. “This study was performed following…” is enough.
- Sample collection: i think this paragraph needs to be restructured in a better order: where, institution for legal permission, the number of samples, the types, and the groups of obtained samples, and the instruments followed by the muscle collection. For example: totoaba individuals were collected from … under/following regulation of Mexican government (mention all institutions). There are three types of habitats…. A total of 142 were obtained between May 2010 - .
- Trace element analysis: muscle and liver preparation, is there any methods reference?
- Line 144-145: last sentence is not necessarily needed, or if needed, put it in the discussion part.
- Statistical analysis: it would be better to move this section after nutritional and risk assessment.

Discussion
- Line 249: the first sentence is not necessary
- Line 281: the standard for Cu level for human health refer to?
- Line 351-353: the sentence “SOD activity can be …” is not necessarily needed.

Figure:
- figure 2 is not needed. SOD analysis was not conducted on this paper. Citing and referencing SOD data in the discussion is possible, but the figure is not necessarily needed.
- Comparison with SOD results, in the discussion section is good, however, please check to make sure if this is allowed or legal to include them in Table 3, Table 4, figure 3. Maybe it is important to check the permission from the journal cited publisher to include this data in the paper.

---

## Round 0.2 · Minor Revisions

Please kindly attend to the minor observations. Looking forward to your revised manuscript

Reviewer 1 ·

Basic reporting

The revised version has taken care of the concerns. I recommend acceptance at this stage.

Experimental design

The revised version has taken care of the concerns. I recommend acceptance at this stage.

Validity of the findings

The revised version has taken care of the concerns. I recommend acceptance at this stage.

Additional comments

The revised version has taken care of the concerns. I recommend acceptance at this stage.

Reviewer 3 ·

Basic reporting

Clear and well described

Experimental design

Well described

Validity of the findings

well described

Additional comments

Thank you very much for the revision. Only some minor comments to clarify:

Materials and Methods
Line 113: please clarify the word “filed” or field?

Results
Line 261-262: 30% of the samples were detected for Cd level. If I get it correct, from Table 3, there are only 14 samples (7 in autumn C, UN and 7 samples in Spring R, MF) 14 out of 144 muscle samples is about 10%. Please clarify.

Line 267: the same, 40% and 70%. There are only 12 samples (Spring R , IF) which were not undetectable out of 144. Please check and clarify

---

## Round 0.3 · accepted · Accept

Thank you very much for addressing all concerns raised by reviewers. The manuscript is now acceptable for publication. Thank you for your fine contribution, and for finding PeerJ as your journal of choice. The editor believes the authors have benefitted greatly from the peer review process to improve the quality of this work. Looking forward to your future submissions to PeerJ.